# Developing an Effective Therapeutic HPV Vaccine to Eradicate Large Tumors by Genetically Fusing Xcl1 and Incorporating IL-9 as Molecular Adjuvants

**DOI:** 10.3390/vaccines13010049

**Published:** 2025-01-09

**Authors:** Zhongjie Sun, Zhongyan Wu, Xuncheng Su

**Affiliations:** 1State Key Laboratory of Elemento-Organic Chemistry, College of Chemistry, Nankai University, Tianjin 300071, China; 2Newish Biological R&D Center, Wuxi 214111, China

**Keywords:** HPV, therapeutic vaccines, Xcl1, IL-9

## Abstract

Background: Human papillomavirus (HPV) is a prevalent infection affecting both men and women, leading to various cytological lesions. Therapeutic vaccines mount a HPV-specific CD8+ cytotoxic T lymphocyte response, thus clearing HPV-infected cells. However, no therapeutic vaccines targeting HPV are currently approved for clinical treatment due to limited efficacy. Our goal is to develop a vaccine that can effectively eliminate tumors caused by HPV. Methods: We genetically fused the chemokine XCL1 with the E6 and E7 proteins of HPV16 to target cDC1 and enhance the vaccine-induced cytotoxic T cell response, ultimately developing a DNA vaccine. Additionally, we screened various interleukins and identified IL-9 as an effective molecular adjuvant for our DNA vaccine. Results: The fusion of Xcl1 significantly improved the quantity and quality of the specific CD8+ T cells. The fusion of Xcl1 also increased immune cell infiltration into the tumor microenvironment. The inclusion of IL-9 significantly elevated the vaccine-induced specific T cell response and enhanced anti-tumor efficacy. IL-9 promotes the formation of central memory T cells. Conclusions: the fusion of Xcl1 and the use of IL-9 as a molecular adjuvant represent promising strategies for vaccine development.

## 1. Introduction

Cervical cancer is predominantly caused by persistent infections with high-risk human papillomavirus (HR-HPV), with approximately 70% of cases attributed to the HPV16 and HPV18 subtypes [1]. Persistent HR-HPV infections can progress to cervical intraepithelial neoplasia (CIN), and about 30% of untreated CIN cases may advance to cervical cancer [2,3,4]. For patients diagnosed with CIN2 or CIN3, current medical guidelines recommend undergoing loop electrosurgical excision procedure (LEEP) resection or cold knife conization. In cases of persistent HR-HPV infection or CIN I, regular follow-up and observation are advised [5]. While surgical interventions are effective for treating CIN2 and CIN3, they pose potential risks to fertility and recurrence post surgery. This highlights a critical limitation in current treatment approaches: there is no effective method for directly eliminating persistent cervical HR-HPV infections, and the management of early lesions often involves a “wait-and-see” approach, which can impose significant psychological stress on patients. Although preventive HPV vaccines are effective in preventing new infections, they do not offer therapeutic benefits for individuals already infected with HPV or those with existing lesions. Consequently, addressing persistent HR-HPV infections and achieving a radical cure for CIN remain unmet medical challenges, underscoring the urgent need for new therapeutic agents to clear persistent cervical HPV infections and treat associated CIN.

Nucleic acid vaccines have the potential to robustly activate the body’s cellular immune response, targeting and eliminating latent HPV-infected and transformed cells. Compared to mRNA vaccines, DNA vaccines offer advantages such as high safety, fewer side effects, cost-effectiveness, good stability, and ease of long-term storage and transportation. Currently, several therapeutic plasmid DNA vaccines targeting the E6 and E7 proteins of HPV16 and HPV18 are under clinical investigation worldwide for the treatment of cervical intraepithelial neoplasia grades 2/3 (CIN2/3). In March 2021, Inovio Corporation announced the results of its Phase III clinical trial, REVEAL 1 (NCT03185013), for the VGX-3100 vaccine. Among 193 subjects with CIN2/3, the vaccination group showed a statistically significant complete lesion resolution and HPV clearance rate of 23.7% (31/131), compared to 11.3% (7/62) in the placebo group. In March 2023, Inovio released results from the REVEAL 2 (NCT03721978) Phase III clinical trial. The vaccination group demonstrated a lesion resolution and virus clearance rate of 27.6% (37/134), significantly higher than the 8.7% (6/69) observed in the placebo group. These findings support VGX-3100 as a safe and effective therapeutic HPV DNA vaccine, potentially becoming the first of its kind to reach the market (available online: https://ir.inovio.com (accessed on 8 January 2025)). To enhance the efficacy of DNA vaccines, researchers explored the addition of cytokine-encoding sequences to the DNA plasmid. This involves fusing immune-enhancing molecules, such as Flt-3L and CCL3, with the E6 and E7 proteins. GX-188E is a DNA vaccine that combines the E6/E7 proteins of HPV16 and HPV18 with Flt-3L. A Phase II single-arm clinical study (NCT02139267) targeting CIN3 showed that 67% (35/52) of patients experienced histopathological regression by the eighth follow-up visit after vaccination [6]. Similarly, VB10.16 is a DNA vaccine that fuses HPV16 E6/E7 proteins with the chemokine CCL3. In a Phase I/IIa clinical trial (NCT02529930) involving 34 women with CIN 2/3, among the 17 evaluable subjects in the extended group, HPV 16 clearance was observed in 47% (8/17) of cases; 94% (16/17) of lesions showed reduction in size, and 59% (10/17) regressed to CIN 1 or normal [7].

However, the main problem is that the receptors of these enhancing molecules are widely distributed in multiple antigen-presenting cell (APC) subpopulations with different functions, resulting in insufficient cytotoxic T cell response and limiting the maximization of their therapeutic effects. Although these vaccines activated a certain degree of cellular immune response in clinical trials, their specific targeting of DC subsets is insufficient, resulting in unsatisfactory clinical therapeutic effects. Mechanistic studies demonstrated that the enhancement of antigen immunogenicity through the fusion of Flt-3L or CCL3 primarily relies on the cDC1 cell subset [8,9]. The cDC1 subgroup garnered significant attention due to its unique ability for antigen cross-presentation. This capability enables the direct activation of cytotoxic lymphocytes, including CD8+ T cells (CTLs), natural killer cells, and NKT cells, positioning cDC1 as a central player in anti-tumor and antiviral immune responses, and thus a key target for enhancing vaccine efficacy [10]. XCR1 is a receptor uniquely expressed in cDC1 cells and is absent in other human cells. As the sole known ligand for XCR1, XCL1 is a critical chemokine with the unique biological function of guiding XCR1-positive cDC1 cells to specific in vivo locations and facilitating their interaction [11,12,13,14,15]. The interaction between XCL1 and its specific receptor XCR1 emerged as a promising area of interest for more effective strategies for therapeutic vaccines. Research has shown that linking the C-terminus of the XCL1 protein to other antigen proteins, such as ovalbumin (OVA) or spike protein, enhances the ability to activate immune responses [16,17]. This finding underscores the potential of XCL1-antigen fusion proteins in the development of more effective immunotherapeutic strategies.

Interleukin-9 (IL-9) is a cytokine that plays a significant role in enhancing cell-mediated immune responses. It is produced by a variety of immune cells, including T helper 9 (Th9) cells, and is involved in promoting the proliferation and survival of various immune cells, such as T cells and mast cells. IL-9 enhances the immune response by modulating the activity of these cells, thereby contributing to the body’s defense against infections and tumors [18]. Additionally, IL-9 can influence the function of other cytokines and immune pathways, further amplifying the immune response. Its role in immune regulation makes it a potential target for therapeutic strategies aimed at boosting immune responses in various diseases [19]. However, the role of IL-9 in nucleic acid vaccines has not yet been studied.

In this study, we developed a therapeutic HPV vaccine in the form of a DNA vaccine, incorporating the chemokine Xcl1 and IL-9 as molecular adjuvants. We evaluated the impact of fusing Xcl1 and adding IL-9 on the immunogenicity of the vaccine.

## 2. Materials and Methods

### 2.1. Cell Lines and Plasmids

The HEK293T cell line (cat. CTCC-001-0188) and the TC-1 (cat. CTCC400-0328) cell line [20], which stably express HPV16 E6E7 protein, were obtained from the Meisen Cell Technology (Jinhua, Zhejiang, China). Cell lines were cultured in DMEM or RPMI1640 medium supplemented with 10% fetal bovine serum, 1% penicillin, and 1% streptomycin. The expression sequences for Xcl1-E6E7, E6E7, mIL-7, mIL-9, mIL-21, and mIL-33 proteins were synthesized and cloned into the pVAX1 vector by Qingke Biotechnology (Beijng, China).

### 2.2. Protein Expression Analysis

HEK293T cells transfected with plasmids were lysed with RIPA lysate (comprising 50 mM Tris HCl, pH 8.0, 150 mM NaCl, 1.5 mM MgCl2, 0.1% SDS, 0.5% deoxycholate (DOC), 1% NP-40, 1 mM PMSF, and a 1×protease inhibitor mixture from Roche, Basel, Switzerland). Following centrifugation at 12,000 rpm for 10 min, the supernatant protein extract was collected and mixed with 5×SDS-PAGE loading buffer, then boiled at 100 °C for 10 min for SDS-PAGE gel analysis. The antibodies used included the HPV16 E7 antibody (Invitrogen, Waltham, MA, USA; cat. MA1-23087), the GAPDH antibody (Biodragon, Suzhou, China; cat. B1034), the P53 antibody (Beyotime Biotechnology, Shanghai, China; cat. AF0255), and the Rb antibody (Santa Cruz Biotechnology, Santa Cruz, CA, USA; cat. sc-102).

### 2.3. Experimental Animals and Immunization

The experimental subjects were 6–8-week-old female C57BL/6 mice, obtained from Charles River (Beijing, China), and housed at the Peking Union-Genius Pharmaceutical Technology Company in Beijing, China. All animal experiments were approved by the Institutional Animal Care and Use Committee (IACUC) of Peking Union-Genius, with the ethical approval number JY-24002. On the day of immunization, designated as Day 0, the mice’s hind limbs were disinfected with alcohol swabs. A sterile syringe was used to inject 25 micrograms of plasmid DNA into the gastrocnemius muscle. Electroporation was then performed using in vivo electroporation equipment from Taresha Biotechnology (Shanghai, China), with parameters set at 60 V, 1 Hz, and 50 ms intervals.

For the deletion of CD4 and CD8 T cell subpopulations and combination therapy with immune checkpoint inhibitors or agonists, the following antibodies were used: Anti-CD4 (BioXcell, Lebanon, NH, USA; cat. BE0119; 350 µg per mouse), Anti-CD8 (BioXcell, Lebanon, NH, USA; cat. BP0061, 250 µg per mouse), Anti-Mouse PD-1 (CD279) (BioXcell, Lebanon, NH, USA; cat. BE0273, 250 µg per mouse), Anti-Mouse CTLA-4 (CD152) (BioXcell, Lebanon, NH, USA; cat. BE0131, 100 µg per mouse), Anti-Mouse PD-L1 (B7-H1) (BioXcell, Lebanon, NH, USA; cat. BP00101, 200 µg per mouse), Anti-4-1BB (BioXcell, Lebanon, NH, USA; cat. BE0239, 350 µg per mouse), and Anti-OX40 (BioXcell, Lebanon, NH, USA; cat. BE0031, 100 µg per mouse). These antibodies were administered intraperitoneally twice a week.

### 2.4. Establishment and Measurement of the TC-1 Xenograft Model

First, the right axillary region of the mice was shaved. Mice were then injected subcutaneously with either 1 × 105 or 5 × 105 TC-1 cells per mouse, depending on the experimental requirements. The mice were randomly assigned to different experimental groups based on tumor volume. Before measuring the tumors, the tumor mass was wiped with an alcohol swab, and the long diameter (L) and short diameter (W) were measured using calipers. Subcutaneous tumors were measured every two days. Tumor volume was calculated using the formula (L × W^2^)/2. Mice were euthanized using carbon dioxide anesthesia when the tumor volume exceeded 2000 mm^3^ or if the tumor showed signs of necrosis. Complete remission was defined as the absence of palpable nodules in two consecutive measurements.

### 2.5. CD11c+CD8+ DC Cells Binding Analysis

Mouse spleens were mechanically disrupted by passing them through a 100 µm sieve (Corning Inc., New York, NY, USA). The resulting suspension was centrifuged at 1600 rpm for 5 min. Red blood cells were lysed using a 1×red blood cell (RBC) lysis buffer (Biolegend, San Diego, CA, USA; cat. 420301) for 2 min, after which the lysis was halted by adding five times the volume of PBS. The suspension was centrifuged again at 1600 rpm for 5 min, and the cells were subsequently counted. For the binding analysis, 1 × 10^7^ mouse spleen cells were incubated with cell lysate obtained from repeated freeze-thaw cycles of HEK293T cells transfected with E6E7 or Xcl1-E6E7 plasmid DNA. The incubation was carried out for 30 min. Following this, the cells were incubated with a FITC-conjugated Flag antibody (Abcam, Cambridge, UK; cat. ab117505) for another 30 min. After washing, the samples were analyzed by flow cytometry using a FACS CANTO TWO flow cytometer (BD Biosciences, San Jose, CA, USA).

### 2.6. E7-Specific T Cell Analysis

Whole blood from mice was collected in PBS containing heparin as an anticoagulant and then centrifuged at 3000 rpm for 5 min. The supernatant was discarded, and the precipitate was gently resuspended. Next, 1×RBC lysis buffer (Biolegend, San Diego, CA, USA; cat. 420301) was added for 1 min at room temperature, followed by centrifugation at 1600 rpm for 5 min. The supernatant was discarded, and the pellet was washed with 1 mL of PBS and centrifuged again at 1600 rpm for 5 min. After discarding the supernatant, the topmost red layer was removed. The cells were then resuspended in 300 microliters of RPMI1640 medium containing 10% FBS. For each sample, 5 microliters of E7_49-57_-specific (MBL Beijing Biotech Beijing, China; cat. TS-5008-1C) tetramer [21] and 1 microliter of anti-CD8 FITC (MBL Beijing Biotech Beijing, China; cat. C299) antibody were added and incubated for 1 h. After washing twice with PBS, the cells can be resuspended for flow cytometry analysis.

### 2.7. Tumor Microenvironment Analysis

Following euthanasia of the mice using carbon dioxide, tumors were excised and finely chopped with scissors. Tumor cell digestion solution was then added, and the mixture was incubated at 37 °C on a shaker for 30 min. The digested cells were filtered through a 100 µm cell strainer to remove large debris, centrifuged at 1600 rpm for 5 min, and washed twice with PBS to obtain a single-cell suspension of tumor tissue. This suspension was used for fluorescence staining and flow cytometry analysis.

For intracellular Granzyme B staining, the BD Fixation/Permeabilization Kit (BD Biosciences, San Jose, CA, USA; cat. 554715) was utilized. The staining protocol for identifying each cell subpopulation was as follows: E7_49-57-specific CD8+ T cells (E7_49-57 tetramer+, CD8+, with or without Granzyme B), NK cells (CD45+ CD3− NK1.1+), CD8+ T cells (CD45+ CD8+), CD4+ T cells (CD45+ CD4+), and Treg cells (CD45+ CD4+ Foxp3+).

Flow cytometric data were acquired using a BD FACS Canto II flow cytometer (BD Biosciences, San Jose, CA, USA) and analyzed with FlowJo software version 7.6.5.

### 2.8. Statistical Analysis

Statistical differences between groups were assessed using a two-tailed *t*-test, with error bars representing the standard error of the mean. GraphPad Prism 5 was utilized for data analysis and plotting, where * *p* < 0.05, ** *p* < 0.005, and *** *p* < 0.001.

## 3. Results

### 3.1. Construction of the HPV Therapeutic Vaccine Fused with Xcl1

Previous research demonstrated that after HPV infection, the E6 and E7 proteins are the principal viral proteins expressed. These proteins interact with tumor suppressor protein P53 and the retinoblastoma protein family member Rb, recruiting E3 ubiquitin ligase to activate the ubiquitin-proteasome pathway. This interaction leads to the degradation of P53 and Rb, disrupting the normal cell cycle, inhibiting apoptosis, accumulating genomic mutations, and facilitating the transformation of infected cells into malignant tumor cells. Consequently, E6 and E7 proteins not only drive cervical epithelial cell carcinogenesis, but also function as antigens specifically expressed by cervical cancer cells [22]. Since E6 and E7 are derived from HPV and are not naturally expressed by the human body, they exhibit low immune tolerance, making them more effective in activating the immune response compared to other human-origin cancer antigens. We selected the E6 and E7 proteins from HPV subtype 16 as linked antigens, given that HPV16 is the subtype responsible for the highest incidence of cervical cancer. To mitigate the oncogenic potential of E6 and E7 expression in nucleic acid vaccines, we mutated their binding sites with P53 and Rb to enhance safety. Based on previous findings, we linked the mutated E6 and E7 proteins end-to-end to form a single entity, referred to as the E6E7 antigen in this text [23]. To preserve the chemotactic properties of Xcl1 and enhance the immunogenicity of the E6E7 antigen, we introduced a flexible linker composed of eleven amino acids (G5SG5) between the chemokine and E6E7 antigen sequences [24]. This hydrophobic amino acid linker is commonly used in fusion proteins to spatially separate the two proteins, ensuring their correct conformational folding and stability. Protein structure predictions were performed using I-TASSER [25] to assess the spatial conformation of the fusion protein. The results indicate that the flexible linker effectively maintained the structural integrity of both the Xcl1 chemokine and the E6E7 antigen (Figure 1A). For ease of detection, we incorporated a FLAG tag into the construct. The FLAG tag sequence was added to the C-terminus of the fusion protein. Given that the animal model used was a mouse, nucleic acid sequences for Xcl1 were derived from mice. The DNA sequences encoding Xcl1, the linker, and the E6E7 antigen protein were optimized for codon usage in mice, resulting in the final DNA sequence for the fusion protein. All DNA sequences were synthesized de novo and cloned into the pVAX1 eukaryotic expression plasmid vector. The plasmids were then amplified, extracted, and transfected into human embryonic kidney (HEK) 293T cells for expression verification. Both the E6E7 and Xcl1-E6E7 plasmids expressed proteins at similar levels in 293T cells, and the levels of P53 and Rb were unaffected, indicating that the mutations effectively prevented their degradation (Figure 1B). We co-incubated cell lysates containing E6E7 and Xcl1-E6E7 proteins with mouse spleen cells. Mouse cDC1 cells were characterized using CD8 and CD11c markers, while the XCR1 antibody was not used to avoid interfering with the binding of Xcl1 to XCR1. A Flag fluorescent antibody was used to indicate the presence of E6E7 or Xcl1-E6E7 proteins on the surface of CD8+CD11c+ cells. The results show that Xcl1-E6E7 bound to more cDC1 cells compared to E6E7, which was similar to the control group (Figure 1C). These findings suggest that the fusion with Xcl1 does not affect protein expression and enhances binding to dendritic cells.

### 3.2. Enhanced Cytotoxic T Lymphocyte (CTL) Activation and Anti-Tumor Efficacy of Xcl1-E6E7

After successfully constructing plasmid DNA vaccines for E6E7 and Xcl1-E6E7, we attempted to investigate the differences in CTL activation induced by varying vaccine dosages. We immunized C57 mice with 5 µg and 25 µg doses of E6E7 and Xcl1-E6E7 plasmid DNA via intramuscular injection combined with electroporation. Post vaccination, we monitored the levels of E7-specific CD8+ T cells in peripheral blood every seven days using E7-specific tetramer. The highest level of E7-specific T cells in the 25 µg dose group was observed on day 14 post injection, followed by a gradual decline. In the low-dose group, peak T cell production was delayed. Notably, Xcl1 fusion significantly boosted E7-specific CD8+ T cell levels in both dosage groups (Figure 2A). Subsequently, we evaluated the vaccines’ tumor-clearing abilities using preventive and therapeutic models.

Initially, we compared T cell responses induced by different vaccination regimens. A single DNA vaccine injection elicited the highest E7-specific CD8+ T cell level on day 14 in mice. Increasing the immunization frequency to two or four within 14 days did not elevate the CD8+ T cell level and sometimes diminished it. Administering a second dose after 14 days failed to reactivate a stronger CD8+ T cell response compared to the initial vaccination (Figure 2B,C). Consequently, we opted for a single-dose regimen in subsequent models. In the prevention model, both 25 µg doses of Xcl1-E6E7 and E6E7 completely inhibited tumor formation. After all the tumors in the control group reached the ethical endpoint, we re-challenged the tumor-free mice from the Xcl1-E6E7 and E6E7 treatment group with a second subcutaneous implantation of TC-1 cells to evaluate the long-term preventive efficacy of the vaccine. The second TC-1 tumor cell inoculation failed to establish tumors, suggesting long-term immunity from a single immunization (Figure 2D). However, at a 5 µg dose, only Xcl1-E6E7 fully prevented tumor formation (Figure 2E).

For therapeutic evaluation, we transplanted TC-1 tumor cells subcutaneously followed by the vaccination. The TC-1 cell line, derived from primary epithelial cells of C57BL/6 mice co-transformed with HPV16 E6 and E7 and c-Ha-ras oncogenes, has been widely used to evaluate the efficacy of vaccines targeting HPV16 E6 and E7 proteins [20]. We initiated vaccination on the fourth day post tumor inoculation, when tumors were not yet visible. A single Xcl1-E6E7 dose on day four post tumor cells inoculation completely eradicated tumors and prevented subsequent tumor formation. In contrast, tumors in the E6E7 group recurred after initial shrinkage (Figure 2F). Accordingly, we measured the levels of specific CD8+ T cell populations in the peripheral blood of tumor-bearing mice after 14 days following vaccine administration. The results demonstrate that the Xcl1-E6E7 consistently maintained elevated levels of specific CD8+ T cells. These findings suggest that the fusion with Xcl1 enhances the immunogenicity of E6E7 and augments its anti-tumor efficacy.

### 3.3. Xcl1-E6E7 Immunotherapy Enhances Antigen-Specific CD8+ T Cell and Innate Immune Cell Infiltration in the Tumor Microenvironment

In order to understand the reasons why the fusion chemokine Xcl1 enhances the anti-tumor activity driven by the E6E7 antigen, we conducted a detailed analysis of specific immune and innate immune cell subpopulations within tumors of mice following vaccine administration. A subcutaneous tumor model was established, and mice were divided into Vector, E6E7, and Xcl1-E6E7 plasmid DNA treatment groups. Fourteen days post DNA vaccine injection, tumors were harvested and processed into single-cell suspensions. Flow cytometry was employed to assess the distribution of E7-specific CD8+ T cells within the tumors across the three groups. In the tumor tissue, where E6E7-specific CD8+ T cells exert their effects, there was a significantly higher proportion of E7-specific CD8+ T cells compared to peripheral blood (Figure 3A). In the E6E7-treated group, levels exceeded 20%, while in the Xcl1-E6E7-treated group, they reached over 40% (Figure 3A). Furthermore, the proportion of E7-specific CD8+ T cells capable of secreting granzyme B was higher in the Xcl1-E6E7 group, indicating a stronger effector state (Figure 3B). The Xcl1-E6E7-treated group also demonstrated a notable increase in the total CD8+ T cell population compared to the E6E7-treated group (Figure 3C).

The full functionality of CD8+ T cells relies on support from other immune subgroups. Therefore, we analyzed changes in other immune cell subpopulations within the tumor. The Xcl1-E6E7 treatment group exhibited a substantial increase in overall white blood cell infiltration compared to the vector control and E6E7 groups, with an average infiltration rate exceeding 50%, whereas the E6E7 group averaged around 30% (Figure 3D). Further analysis of CD4+ T cells and Treg cell proportions revealed that the total CD4 also increased, but Treg did not increase and showed a certain downward trend, although there was no significant difference, suggesting that the TC-1 transplanted tumors did not modulate Xcl1-E6E7 activity through Treg enrichment (Figure 3E,F). Natural immune cells also play an important role in anti-tumor immunity, mainly by promoting or inhibiting anti-tumor immune responses [26]. We primarily analyzed NK cell subpopulations, and the results indicate that the proportion of NK cells in the Xcl1-E6E7 treatment group was significantly higher than in the E6E7 group (Figure 3G). This suggests that Xcl1-E6E7 facilitates the infiltration of both specific and natural effector cell subpopulations into the tumor microenvironment.

### 3.4. IL-9 as a Molecular Adjuvant Enhances the Cellular Immune Response and Anti-Tumor Efficacy of Xcl1-E6E7

As shown in Figure 2, we analyzed the therapeutic effects of the vaccine prior to tumor formation, considering the clinical progression from CIN to cervical cancer, where tumor burden gradually increases. We then initiated vaccination on the seventh day, when average tumor volume reached approximately 30 mm^3^, to simulate higher tumor burdens and assess the efficacy of the vaccine under this condition. In cases of established tumors, while a single Xcl1-E6E7 dose achieved tumor suppression, it was not curative, with recurrences observed, whereas the E6E7 group exhibited even weaker anti-tumor effects under this condition (Figure 4A–C). We analyzed and compared the levels of specific T cells induced by Xcl1-E6E7 under varying tumor loads and found that higher tumor burdens were associated with lower levels of specific T cell induction (Figure 4D). This inverse relationship may partly explain the diminished therapeutic effect observed in established tumors.

The use of interleukins as molecular adjuvants in vaccines to boost immune responses has been extensively studied [27]. To further enhance the anti-tumor efficacy of Xcl1-E6E7, we explored the immunization of mice with the Xcl1-E6E7 plasmid in combination with several plasmids expressing interleukins known to stimulate T cell proliferation and memory formation, including IL-7, IL-9, IL-21, and IL-33. Our results demonstrate that the combination with IL-9 induced the highest levels of E7-specific CD8+ T cells (Figure 4E). To determine whether this effect was specific to Xcl1-E6E7, we also constructed a plasmid expressing mouse-derived GPC3 (amino acids 1–553) with a C-terminal FLAG-tag fusion. We also co-immunized mice with a GPC3 plasmid and the IL-9 plasmid. IL-9 similarly enhanced the GPC3-specific cellular immune response, suggesting that this enhancement has a degree of universal applicability (Figure 4F,G).

We further analyzed the phenotype of specific CD8+ T cells induced by Xcl1-E6E7, with or without IL-9. Changes in CD8+ memory precursor effector cells (MPECs, CD127+ KLRG1−), short-lived effector cells (SLECs, CD127− KLRG1+), early effector cells (CD127− KLRG1−), and double-positive cells (CD127+ KLRG1+) were evaluated using CD127 and KLRG1 expression as markers. The results reveal that the inclusion of IL-9 not only increased the proportion of MPECs of specific CD8+ T cells, but also promoted the differentiation of central memory CD8+ T cells, thereby improving the quality of the specific T cell response (Figure 4H,I). We then evaluated the anti-tumor efficacy of combining Xcl1-E6E7 with the IL-9 plasmid. When the average tumor volume reached 30 mm^3^, mice were grouped to receive either Xcl1-E6E7 with or without the IL-9 plasmid. The results demonstrate that mice treated with the combination of Xcl1-E6E7 and IL-9 experienced earlier tumor regression and ultimately achieved a complete remission rate of 3 out of 5, compared to a 0 out of 5 remission rate for those treated with Xcl1-E6E7 alone (Figure 4J). These findings suggest that IL-9 significantly enhances the vaccine-induced specific T cell immune response and promotes the formation of immune memory.

### 3.5. The Xcl1-E6E7+mIL-9 Plasmid DNA Vaccine, When Used in Combination with a CTLA-4 Inhibitor, Has Shown Potential to Eradicate Established HPV+ Tumors

We further investigated whether the effectiveness of Xcl1-E6E7+mIL-9 depends on CD8+ T lymphocytes. To explore this, antibodies were used to deplete CD4+ or CD8+ T cells, allowing observation of the preventive effect of Xcl1-E6E7+mIL-9. Seven days post immunization, antibodies targeting CD4+ or CD8+ T cells were administered intraperitoneally twice a week. One week after antibody administration, effective depletion of these T cells was confirmed (Figure 5A). Subsequently, the TC-1 tumor cell line was inoculated 14 days after vaccination to assess tumor formation. Results indicate that Xcl1-E6E7+mIL-9 maintained its function following CD4+ T cell depletion but failed to prevent tumor development after CD8+ T cell depletion (Figure 5B).

Immune checkpoint inhibitors (ICIs) are a class of drugs that enhance the immune system’s ability to recognize and destroy tumor cells by blocking inhibitory pathways. Under normal physiological conditions, the immune system employs checkpoints to regulate T cell activity, thereby preventing autoimmune reactions [28]. Notable checkpoints successfully applied in clinical trials include the PD-1/PD-L1 and CTLA-4 pathways [29,30]. Despite significant advancements in cancer treatment with ICIs, not all patients benefit, and some may not respond at all. A key challenge to ICI efficacy is the insufficient infiltration of immune cells into tumors. However, this study demonstrates that the Xcl1-E6E7 vaccine can induce infiltration by various natural and adaptive immune cells, potentially improving ICI response rates.

To explore this potential, the study combines Xcl1-E6E7+mIL-9 with PD-1, PD-L1, and CTLA-4 inhibitors, along with two immune agonists, OX40 and 4-1BB. To fully demonstrate the benefits of combination therapy, we first evaluated the therapeutic effects of different doses of the Xcl1-E6E7+mIL-9 vaccine on tumors. Our findings indicate that a 25 µg dose of the Xcl1-E6E7+mIL-9 vaccine is sufficient to completely eradicate subcutaneous TC-1 tumors, which achieved complete tumor clearance in about 60% of the mice (Figure 5C). At this dose, the added benefit of combining with ICs might not be apparent. In contrast, the therapeutic efficacy of the 10 µg dose of the Xcl1-E6E7+mIL-9 plasmid DNA vaccine was notably reduced compared to the 25 µg dose, with none of the five mice experiencing complete tumor remission (Figure 5C). Consequently, the study reduces the Xcl1-E6E7+mIL-9 plasmid DNA vaccine dose to 10 µg to thoroughly assess its synergistic therapeutic effect with OX40 and 4-1BB agonists and PD-1, PD-L1, and CTLA-4 inhibitors. Once the average tumor volume of TC-1 reached approximately 75 mm^3^, mice were randomly assigned to the following groups: vehicle control, OX40 and 4-1BB agonists, PD-1, PD-L1, and CTLA-4 inhibitors, Xcl1-E6E7+mIL-9 plasmid DNA vaccine monotherapy, and Xcl1-E6E7+mIL-9 plasmid DNA immunization combined with OX40 and 4-1BB agonists and PD-1, PD-L1, and CTLA-4 inhibitors. The five checkpoint drugs were administered via intraperitoneal injection every three days. Tumor volume was measured every other day to construct and analyze the tumor growth curves in mice.

Results indicate that among the five checkpoint drugs, only the 4-1BB agonist monotherapy demonstrated a degree of treatment-mediated tumor growth inhibition (Figure 5D). In the combination therapy group with the CTLA-4 inhibitor, complete tumor regression was observed in 3 out of 5 mice. Tumor shrinkage began earlier compared to the Xcl1-E6E7+mIL-9 plasmid DNA vaccine monotherapy group, and even when recurrence occurred, it was delayed (Figure 5E). However, no significant benefit was observed when combined with other immune checkpoint drugs. Furthermore, the study analyzed E7-specific CD8+ T cells in the peripheral blood of mice treated with the CTLA-4 inhibitor. The combination therapy with CTLA-4 inhibitors significantly upregulated the levels of E7-specific CD8+ T cells (Figure 5F). These results suggest that combining CTLA-4 inhibitors with the Xcl1-E6E7+mIL-9 plasmid DNA vaccine is particularly effective in enhancing therapeutic outcomes.

## 4. Discussion

Over the past few decades, the administration of prophylactic HPV vaccines significantly reduced the incidence of cervical cancer. However, individuals already infected with HPV remain a key group at risk for developing cervical cancer. Currently, there are no effective treatments for this population. The standard approach involves regular monitoring and surgery when necessary. However, surgery does not eliminate HPV, resulting in a high risk of recurrence. The transformation of cells caused by HPV infection is dependent on the E6 and E7 proteins. Therapeutic DNA vaccines targeting these proteins completed Phase III clinical trials and have shown some therapeutic benefits, though the proportion of patients benefiting is less than 30%. Our research indicates that as tumor burden increases, the cell-mediated immune response induced by the vaccine is also weakened, highlighting the importance of further improving the immunogenicity of existing vaccines. In this study, we genetically fused Xcl1 and combined IL-9 as adjuvants, significantly enhancing the vaccine’s immunogenicity. This novel vaccine effectively cleared large established tumors that could not be eliminated by targeting E6 and E7 alone. Additionally, when combined with a CTLA-4 inhibitor, it demonstrated superior antitumor activity. Therefore, our research provides a basis for improving and developing more effective therapeutic HPV vaccines.

Interleukin-9 (IL-9) is a cytokine produced by various cell types, including TH2, TH9, TH17, regulatory T (Treg) cells, mast cells, type 2 innate lymphoid cells (ILC2), and CD8+ T cells. IL-9 activates the phosphorylation of Janus kinase 1 (JAK1) and JAK3 by binding to its receptor, IL-9R, thus initiating downstream transcriptional signaling and exerting its biological functions. Similar to IL-9, the IL-9 receptor is also widely expressed across different cell types, which allows IL-9 to perform a broad range of regulatory functions. These include acting as a growth factor to promote T helper cell proliferation, stimulating mast cell proliferation and activation, and activating Treg cells to secrete transforming growth factor-beta (TGF-β) [18]. The role of IL-9 in tumor development and progression is complex due to its diverse functions, and it varies depending on the type of cancer. In lymphoma, tumor cells often secrete IL-9 to promote their own proliferation and inhibit apoptosis. Similarly, in lung and breast cancers, TH9 cells contribute to cancer progression by secreting IL-9. Conversely, the adoptive transfer of TH9 cells in melanoma demonstrated therapeutic effects similar to those of chimeric antigen receptor T (CAR-T) cell therapy. Additionally, IL-9 enhances anti-tumor immunity by activating mast cells and indirectly boosting the cytotoxic functions of CD8+ T cells and natural killer (NK) cells [31]. Despite these findings, the direct impact of IL-9 on vaccine-induced immunogenicity remains unexplored. Our research indicates that IL-9 not only increases the number of antigen-specific T cells, but also promotes the formation of immune memory. This effect may be attributed to IL-9’s ability to inhibit apoptosis and facilitate immune memory formation, or it could be due to IL-9’s role in recruiting mast cells and eosinophils, thereby enhancing antigen-driven immune activation. These hypotheses require further experimental validation, which is a limitation of our current study.

We explored the synergistic effects of various immune checkpoint drugs in combination with our DNA vaccine. Notably, the combination of the Xcl1-E6E7+mIL-9 DNA vaccine with a CTLA-4 inhibitor yielded the most effective therapeutic outcome. This combination significantly enhanced the levels of E7-specific T cells induced by the DNA vaccine. CTLA-4 competes with the co-stimulatory receptor CD28 for CD80 and CD86 ligands, thereby inhibiting T cell activation. The release of negative co-stimulatory signals mediated by CTLA-4, which enhances effector T cell-mediated immune responses, is recognized as the core mechanism by which CTLA-4 inhibitors exert their effects [32,33,34]. Additionally, regulatory T cells (Tregs) express high levels of CTLA-4 on their surface. Several murine studies suggest that CTLA-4 inhibitors may reduce Treg-mediated suppression and eliminate Tregs in the tumor microenvironment (TME) through antibody-dependent cell-mediated cytotoxicity (ADCC). This leads to decreased tumor immune suppression and an expansion of effector T cells, thereby promoting an anti-tumor immune response [35,36]. In this study, CTLA-4 inhibitors were observed to increase the number of antigen-specific effector T cells in peripheral blood. While several vaccines have shown efficacy in combination with PD-1 inhibitors, which function by reversing the exhausted T cell phenotype, the combination with PD-1 inhibitors in this study did not enhance the vaccine’s anti-tumor effect. To understand why the combination of the Xcl1-E6E7+mIL-9 DNA vaccine and PD-1 inhibitor is ineffective, further investigation is required to assess whether the tumor-specific T cells induced by the DNA vaccine are in a PD-1-responsive state. The findings of this study highlight the importance of experimental validation before combining various immune checkpoint therapies with different vaccine platforms.

In our study, we utilized subcutaneous xenografts derived from the TC-1 cell line to assess the efficacy of our DNA vaccine. However, this model is highly sensitive to therapeutic vaccines, and several vaccines that demonstrated effectiveness in this model have not translated well clinically [23,37,38,39]. This discrepancy may be attributed to the high levels of E6 and E7 proteins in the TC-1 cell line, which likely exceed those in actual HPV-induced cancer cells, allowing for more efficient presentation on the cell surface for immune recognition and destruction. Furthermore, since mice have not been previously exposed to HPV, there is no pre-existing immune tolerance to E6 and E7, which facilitates a strong immune response to the vaccine. In contrast, the progression to cervical intraepithelial neoplasia in humans typically results from long-term HPV infection, which involves a degree of immune tolerance. To better replicate human HPV infection, it would be advantageous to develop E6 and E7 transgenic mice for evaluating vaccine efficacy.

## 5. Conclusions

By fusing the E6 and E7 protein with Xcl1 to target cDC1, we developed a DNA vaccine capable of inducing a stronger cytotoxic T lymphocyte immune response. Its therapeutic efficacy can be further enhanced by the combination with an IL-9 molecular adjuvant.

## Figures and Tables

**Figure 1 vaccines-13-00049-f001:**
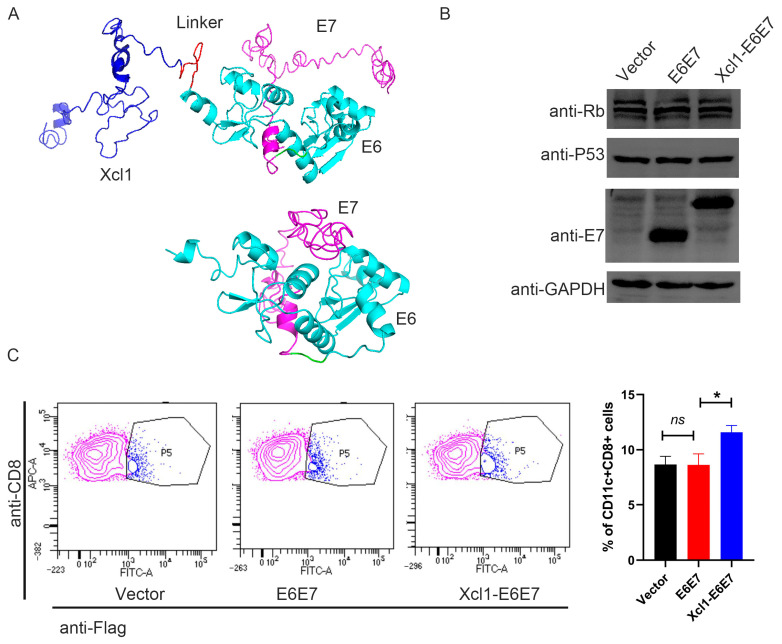
Construction and analysis of HPV16 therapeutic DNA vaccine fused with chemokine Xcl1. (**A**) The structural prediction diagram of the E6E7 protein and the Xcl1-E6E7 fusion protein. In the diagram, dark blue indicates Xcl1, red represents the linker, bright blue denotes E6, and light purple signifies E7. (**B**) Following the transfection of HEK293T cells for 24 h, Western blot analysis was conducted to assess the expression levels of the E6E7 and Xcl1-E6E7 DNA vaccines, alongside the protein levels of P53 and Rb. (**C**) Flow cytometry was employed to determine the proportion of CD11c+CD8+ DC cells bound to E6E7 and Xcl1-E6E7 proteins. The left panel presents a representative flow cytometry chart, while the right panel displays the corresponding statistical analysis (N = 3, mean ± SEM).

**Figure 2 vaccines-13-00049-f002:**
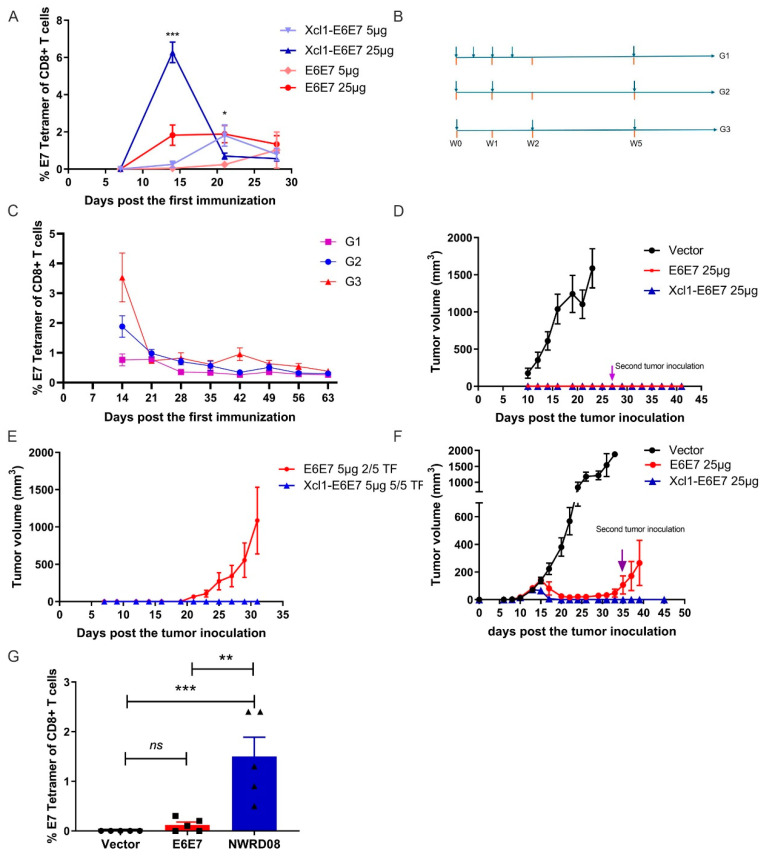
Efficacy analysis of E6E7 and Xcl1-E6E7 plasmid DNA vaccines. (**A**) E7-Specific CD8+ T cell response. Peripheral blood samples were collected on days 9, 14, 21, and 28 following the administration of 5 µg and 25 µg doses of E6E7 and Xcl1-E6E7 vaccines. The E7-specific CD8+ T cell responses were dynamically analyzed (N = 5, mean ± SEM). (**B**) The DNA vaccine injection frequency patterns. (**C**) The graph illustrates the dynamics of E7-specific T cell levels over time under various injection frequency regimens (N = 5, mean ± SEM). (**D**,**E**) After 14 days later administering 25 µg (**D**) and 5 µg (**E**) doses of the E6E7 and Xcl1-E6E7 vaccines, tumor cells were inoculated to assess tumor growth, with growth curves plotted accordingly (N = 5, mean ± SEM). (**F**) Tumor growth post inoculation and vaccination. Four days following tumor cell inoculation, 25 µg doses of the E6E7 and Xcl1-E6E7 vaccines were administered. Tumor sizes were measured and growth curves were plotted. A second tumor inoculation was conducted in mice where tumors completely regressed (N = 5, mean ± SEM). (**G**) Fourteen days post vaccination in figure (**F**), flow cytometry was used to assess E7-specific CD8+ T cell levels in the peripheral blood of mice from each group (N = 5, mean ± SEM).

**Figure 3 vaccines-13-00049-f003:**
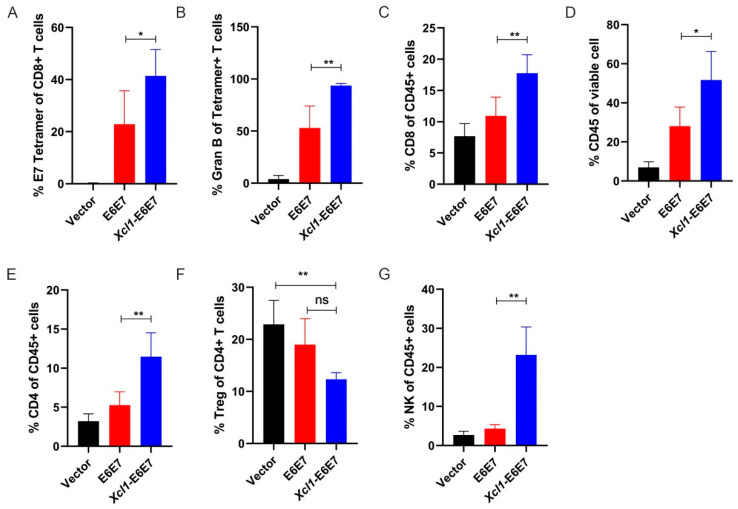
Analysis of immune cell subpopulations within the tumor microenvironment. (**A**) Flow cytometry analysis of the proportion of E7-specific CD8+ T cells in tumors from mice treated with E6E7 or Xcl1-E6E7 vaccines. (**B**) Detection of the proportion of Granzyme B (GranB)-positive cells within the E7-specific CD8+ T cell population in each treatment group. (**C**–**E**) Flow cytometry analysis of CD45+ total white blood cells and the proportions of CD8+ and CD4+ T cells within the tumor microenvironment of mice treated with E6E7 or Xcl1-E6E7 vaccines. (**F**) Flow cytometry analysis of the proportion of regulatory T cells (Tregs) within the CD4+ T cell population. (**G**) Flow cytometry analysis of the proportion of natural killer (NK) cells relative to total white blood cells in the tumor microenvironment of mice in each group. (N = 5, mean ± SEM) for each figure.

**Figure 4 vaccines-13-00049-f004:**
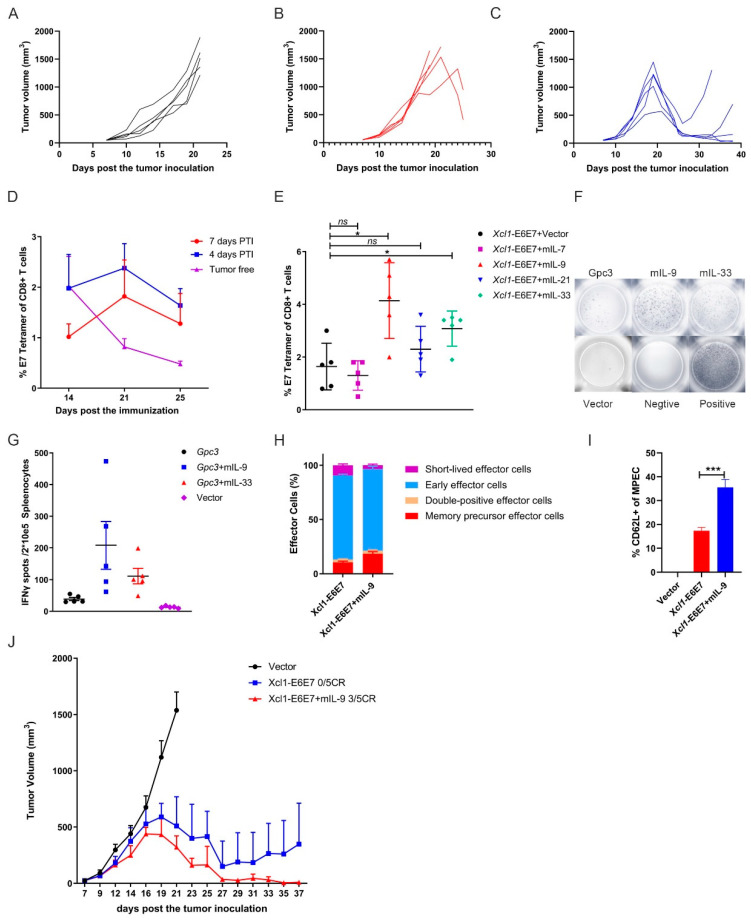
IL-9 enhances the therapeutic effect of the Xcl1-E6E7 plasmid DNA vaccine. (**A**–**C**) Mice with established subcutaneous tumors, averaging 30 mm^3^ in volume, formed by 1 × 10^5^ TC-1 cells per mouse, were treated with either E6E7 or Xcl1-E6E7 plasmid DNA. Tumor growth curves were plotted for individual mice across the empty vector group (**A**), E6E7 group (**B**), and Xcl1-E6E7 group (**C**). (**D**) Xcl1-E6E7 DNA vaccine was administered 4 or 7 days after TC-1 tumor cell inoculation. The levels of E7-specific CD8+ T cells in peripheral blood were dynamically monitored and plotted as a curve. (**E**) Fourteen days post immunization with the Xcl1-E6E7 plasmid DNA vaccine combined with plasmid DNA expressing IL-7, IL-9, IL-21, or IL-33, E7-specific CD8+ T cell levels were assessed in peripheral blood. (**F**,**G**) Fourteen days after immunization with plasmid DNA expressing IL-9 and IL-33 in combination with GPC3 plasmid DNA vaccines, ELISPOT analysis of mouse spleen cells was conducted (**G**), F showing representative IFN-γ spots. (**H**,**I**) The MPEC proportion of E7-specific CD8+ T cells in the spleen of Xcl1-E6E7 with and without IL-9-immunized mice was analyzed after 28 days, along with the expression of the central memory marker CD62L in MPECs. (**J**) The plots show the tumor growth curve of Xcl1-E6E7 with and without IL-9-treated mice (N = 5, mean ± SEM).

**Figure 5 vaccines-13-00049-f005:**
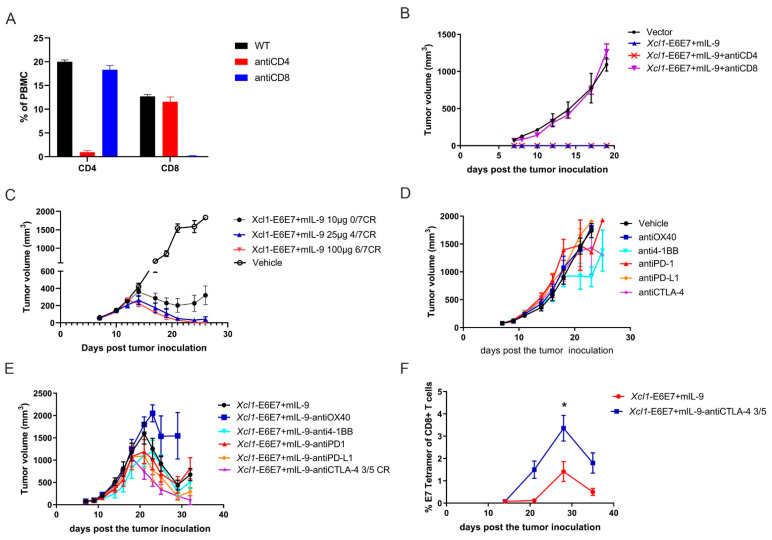
Efficacy analysis of Xcl1-E6E7 + mIL-9 combined with immune checkpoint inhibitors. (**A**) Flow cytometry analysis of the proportions of CD4+ and CD8+ T cells in peripheral blood, conducted one week after the administration of anti-CD4 and anti-CD8 antibodies. (**B**) Fourteen days post immunization with Xcl1-E6E7 + mIL-9, mice were inoculated with 5 × 10^5^ TC-1 tumor cells to observe tumor growth. Anti-CD4 and anti-CD8 antibodies were administered intraperitoneally starting one week after vaccination, with dosing twice a week (N = 5, mean ± SEM). (**C**) Mice with TC-1 xenograft tumors were treated with 10, 25, and 100 µg doses of the Xcl1-E6E7+mIL-9 vaccine. Tumor volumes in each group were measured regularly, and growth curves were plotted (N = 7, mean ± SEM). (**D**,**E**) Mice were subcutaneously inoculated with 5 × 10^5^ TC-1 tumor cells to form tumors, then randomly assigned to treatment groups. Treatments included anti-OX40, anti-4-1BB, anti-PD-1, anti-PD-L1, and anti-CTLA-4, either alone (**D**) or in combination with Xcl1-E6E7 + mIL-9 plasmid DNA (**E**). Tumor growth was monitored, and growth curves were plotted (N = 5, mean ± SEM). (**F**) Dynamic monitoring of E7-specific CD8+ T cell levels in the peripheral blood of mice immunized with XCL1-E6E7 + mIL-9 plasmid DNA, either alone or in combination with anti-CTLA-4 antibody (N = 5, mean ± SEM).

## Data Availability

The original contributions presented in this study are included in the article. Further inquiries can be directed to the corresponding author.

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
