# Peer review of "Developing an Effective Therapeutic HPV Vaccine to Eradicate Large Tumors by Genetically Fusing Xcl1 and Incorporating IL-9 as Molecular Adjuvants"

_vaccines, 2025, doi:10.3390/vaccines13010049_

Round 1

Reviewer 1 Report

Comments and Suggestions for Authors

The authors of Developing an effective therapeutic HPV vaccine to eradicate large tumors by genetically fusing Xcl1 and incorporating IL-9 as molecular adjuvants paper, are presenting a very comprehensive study in which they developed a therapeutic HPV vaccine in the form of a DNA vaccine, incorporating the chemokine Xcl1 and IL-9 as molecular adjuvants. Also, they evaluated the impact of fusing Xcl1 and adding IL-9 on the immunogenicity of the vaccine.

As materials and methods have been used Cell Lines and Plasmids, Protein Expression Analysis, Experimental Animals and Immunization, CD11c⁺CD8⁺ DC cells Binding Analysis, E7-Specific T Cell Analysis, and Tumor Microenvironment Analysis.

The Results are sustained by 5 figures:

Figure 1. Construction and Analysis of HPV16 Therapeutic DNA Vaccine Fused with Chemokine 223 Xcl1.

Figure 2. Efficacy Analysis of E6E7 and Xcl1-E6E7 Plasmid DNA Vaccines.

Figure 3. Analysis of Immune Cell Subpopulations Within the Tumor Microenvironment.

Figure 4. IL-9 Enhances the Therapeutic Effect of the Xcl1-E6E7 Plasmid DNA Vaccine.

Figure 5. Efficacy Analysis of Xcl1-E6E7 + mIL-9 Combined with Immune Checkpoint Inhibitors.

The authors present the limitations of their study.

One question: were the cell lines experiments performed duplicate?

Reviewer 2 Report

Comments and Suggestions for Authors

The manuscript by Sun et al. entitled “Developing an effective therapeutic HPV vaccine to eradicate large tumors by genetically fusing Xcl1 and incorporating IL-9 as molecular adjuvants (vaccines-3393515)” described results examining DNA vaccine encoding Xcl-1-E6-E7 tandemly linked fusion protein using TC-1 cell xenograft model for therapeutic use to cancer caused by HPV. Having Xcl-1, with HPV E6 and E7, would guide the linked protein to conventional dendritic cells for its cross presentation and subsequent activation of CD8+ T cells, which were expected to play a key role on controlling the growth of cancer likely expressing E6 and E7. They showed that vaccinating with Xcl-1-E6-E7 expressing DNA was more efficient in controlling tumor growth in preventive and therapeutic set-up, and co-vaccination with mIL-9 expressing DNA was more effective than vaccinating Xcl-1-E6-E7 alone. Further combination with an immune check point inhibitor, CTLA-4 inhibitor, appeared to be the most effective among the tested conditions in preventing tumor growth and occasionally led to complete remission in therapeutic setting.

They presented apparently impressive set of data to address their points, though some details were somewhat confusing and needed modifications. Descriptions of dose and timing of vaccination, which is vital to evaluate the effectiveness of the regimen, were somewhat incomplete in this manuscript. The major drawback of this manuscript was use of TC-1 xenograft model. It is well known that this model is highly responsive to therapeutic vaccine challenges, but those proved effective in this model were often found ineffective in clinical challenges, which should be mentioned in discussion section.

a) Regimens, including methods and timings of TC-1 cell inoculation, should be elaborated.

1.         The origin of TC-1 cells is not clear. It simply described purchased vender, but it should be explained if this is common TC-1 widely used as xenograft model for cancer caused by expression of HPV E6 and E7.

2.         Sometimes the timing of DNA vaccination was set by days after TC-1 inoculation, but others set by volume of the tumor reaching 30mm3 (line 366) or 75mm3 (line 414). It was not clear if the 8th days after TC-1 inoculation equals the day when the tumor volume reaching 30 or 50mm3 (In lines 270-271 it was described that average tumor volume would reach 50mm3 on 8th days post-inoculation, but it was said 30mm3 in lines 326-327). It is even unclear what was the days after inoculation when tumor volume reached 75mm3. If the timing of DNA vaccination was not set by days after the inoculation, methods to estimate tumor volume in mice should be elaborated.

3.         Similarly, timing of TC-1 inoculation was not clear in CD4+/CD8+ T cell depletion experiment (lines 377-381, Fig. 5A).  

4.         In the Fig. 5C, D, and E, the vaccine dose was reduced from 25 to 10ug. Although dose dependency was incompletely examined, why 10ug was good enough to “ thoroughly assess its synergistically effect”? Similarly, what was the basis of 75mm3 tumor volume was selected for the evaluation? Any data in hand to support this approach?

5.         The methods to measure tumor volume was not shown. What was the criteria of complete remission?

b) Information on experimental materials and methods was missing.

1.         Information of Flag-tag location in Xcl-1-E6-E7 fusion gene (lines 204-205) and of GPC3 expressing plasmid (lines 358-359) was missing.

2.         It could be a standardized method, but method to detect “E7 tetramer…” better be described or with reference in elsewhere.

3.         Methods extracting data shown in Fig. 4H should be elaborated.

c) Importance of combining ICI, addressed in the results shown in Fig. 5D, D, and E, should be included in discussion section.

d) In Fig .2 the descriptions in the legend and text did not match.

1.         In the text it was described that effect of vaccination was examined 4th days and 8th days post TC-1 inoculation (lines 269-273), but only the data of the 4th days vaccination was shown (Fig.2F).

2.         In Fig. 2D there was an indication “second inoculation”, which was not explained in the text.

Round 2

Reviewer 2 Report

Comments and Suggestions for Authors

The revised manuscript by Sun et al. (vaccines-3393515) examined therapeutic DNA vaccine encoding Xcl-1-E6-E7 tandemly linked fusion protein using TC-1 cell xenograft model targeting cancer caused by HPV infection. The authors have properly addressed the points raised by the reviewer, and the text has been significantly improved after the revision. This reviewer feels the revised manuscript meets the standards for publication in Vaccines.